# DFT Investigations of the Vibrational Spectra and Translational Modes of Ice II

**DOI:** 10.3390/molecules24173135

**Published:** 2019-08-28

**Authors:** Jing-Wen Cao, Jia-Yi Chen, Xiao-Ling Qin, Xu-Liang Zhu, Lu Jiang, Yue Gu, Xu-Hao Yu, Peng Zhang

**Affiliations:** School of Space Science and Physics, Shandong University, Weihai 264209, China

**Keywords:** ice II, translational modes, hydrogen bond, CASTEP, first principles, density functional theory

## Abstract

The vibrational spectrum of ice II was investigated using the CASTEP code based on first-principles density functional theory (DFT). Based on good agreement with inelastic neutron scattering (INS), infrared (IR), and Raman experimental data, we discuss the translation, libration, bending, and stretching band using normal modes analysis method. In the translation band, we found that the four-bond and two-bond molecular vibration modes constitute three main peaks in accordance with INS ranging from 117 to 318 cm^−1^. We also discovered that the lower frequencies are cluster vibrations that may overlap with acoustic phonons. Whale et al. found in ice XV that some intramolecular vibrational modes include many isolated-molecule stretches of only one O–H bond, whereas the other O–H bond does not vibrate. This phenomenon is very common in ice II, and we attribute it to local tetrahedral deformation. The pathway of combining normal mode analysis with experimental spectra leads to scientific assignments.

## 1. Introduction

Ice, or solid water, is the most polymorphic single-component molecular substance, and it shows great versatility and unique properties [1]. More than 17 different phases have been experimentally established so far [2]. The hydrogen bonds (H-bonds) form the bridge network that binds water and other substances. However, many mechanisms of H-bond interactions in water remain unknown. For ice Ih, it is widely accepted that the peak at ~230 cm^−1^ seen in vibration spectra by IR absorption [3] and Raman scattering [4] is the intermolecular H-bond vibration frequency. In 1989, a high-resolution inelastic neutron scattering (INS) experiment found two distinct peaks at 28.2 and 37.7 meV (i.e., 227 and 304 cm^−1^) in the translation band [5]. Later, Li et al. found that the two main peaks in similar position exist in many ice phases [6]. This led to the question of where the strong peak comes from. Our previous investigations on ice Ic, XIV, XVI, XVII, and VII/VIII revealed that the two peaks are from two intrinsic vibrational modes in the ice lattice [2,7,8,9,10]. Herein, we analyze the normal modes, especially in the terahertz region of the high-pressure phase of ice II. We found that the two categories of vibrations constitute three main peaks in accordance with INS. Furthermore, we revealed that the much lower optic modes are from cluster vibrations.

Ice II has an ordered hydrogen arrangement that was first discovered by Tammann in 1900 [11,12]. In 1964, Kamb first performed a structural analysis of ice II by single-crystal X-ray diffraction, which yielded evidence that the hydrogens in ice II are ordered [13]. Finch et al. used neutron diffraction to verify the proposed ordering scheme [14,15]. Dielectric measurements by Wilson et al. and entropy measurements by Whalley et al. confirmed the existence of orientational order throughout the stable area of ice II [16,17].

At a pressure of 0.2 GPa, ordinary hexagonal ice Ih leads to the formation of ice II. When compressed beyond 0.5 GPa, ice II transforms into ice V or ice VI, which are both denser polymorphs. Ice II does not melt into liquid water; it changes into either ice III, V, or VI when heated above 240 K [11]. The current standard method used to produce ice II is the isothermal compression of hexagonal ice Ih at about 200 K. Ice II can also be produced by the decompression or cooling of ice V [18,19]. Lobban et al. first successfully investigated the reliable structural data of ice II obtained under pressure [20]. The structure of ice II is based on a rhombohedral cell, space group $R\bar{3}$, whose unit cell dimensions are a = 7.78 Å and α = 113.1° [13]. It can also be described with aH = 12.92° and cH = 6.23 Å for the corresponding hexagonal cell [20]. The density of ice II is 1.17 g/cm^3^ at T = 123 K and ambient pressure [13]. In 2018, Shephard et al. exposed the strict topologically constrained nature of its H-bond network [21].

The IR spectra of ice II were investigated in the 1960s [22,23], and Raman scattering and INS experiments were reported later [24,25,26]. However, reports of theoretical assignments are lacking. In this work, we analyze the molecular and atomic vibrational modes. The “isolated stretching mode” found by Whale et al. is also discussed in this work.

## 2. Computational Methods

Using the CASTEP [27] code based on the first-principles density functional theory (DFT) method, the phonon density of states (PDOS) of ice II were calculated. We chose the RPBE [28] exchange-correlation functional of the generalized gradient approximation (GGA) according to our test, which produced a slight redshift in the translation band but had acceptable accuracy in the stretching band [29]. The energy threshold and self-consistent field tolerance were set as 1 × 10^−9^ eV/atom for geometry optimization. The energy cutoff was 750 eV and the k-point was 2 × 2 × 2. Norm-conserving pseudopotentials were used to calculate the PDOS by the linear response method. Since the pressure of laboratory preparation was from 0.2 to 0.5 GPa, we tested 0.2, 0.3, 0.4, 0.5, and 1.0 GPa for geometry optimization, respectively. According to the experiment, the highest intensity of the Raman peak was at 3189 cm^−1^ [24]. Based on this benchmark, we selected the best matchable data, 0.5 GPa, for comparison with the experimental results hereinafter. (Please see the comparisons of normal modes between 0.3, 0.5, and 1.0 GPa in Appendix A).

## 3. Results and Discussion

The computing spectra of Raman scattering, IR absorption, and PDOS are shown in Figure 1, which is divided into four parts according to the four separate vibrational regions. Due to the wide scale of the intensity in different regions, we adjusted their proportions for comparison. Since the INS may collect phonon signals throughout the Brillouin zone (BZ), there are qualitative similarities between the INS spectra and PDOS. However, photon scattering/absorption can only detect the signals near the BZ center. Comparisons between PDOS and INS, and between normal modes and IR/Raman peaks, are shown in Table 1. A primitive cell contains 12 molecules; thus, there are 12 × 3 × 3 − 3 = 105 optical normal modes. The related typical normal modes are discussed later in this paper.

When it comes to the translation band, the curve detected by INS includes the phonons of acoustic modes, cluster vibrational modes, and monomer modes in many ice phases. The Raman and IR peaks are from optic modes near the BZ center that are detectable, subject to selection rules. Based on harmonic approximation, there are 33 optic normal modes from 54 to 318 cm^−1^. Note that the peaks are matched with normal modes by the calculated intensities. The Raman and IR peaks may correspond to different normal modes subject to different selection rules. Bertie et al. reported seven IR peaks at 107, 136, 151, 186, 253, 295, and 335 cm^−1^ [23]. Compared with the normal modes, the corresponding wavenumbers are 130, 157, 164, 206, 223, 293, and 318 cm^−1^. Raman spectroscopy detected eight peaks at 72, 104, 152, 187, 200, 262, 268, and 322 cm^−1^ [25], of which the first seven matched the normal modes of 90, 117, 154, 176, 219, 262, and 298 cm^−1^. Since there are many normal modes in each band and the experimental data are rare, we matched a Raman or IR active normal mode with the biggest intensity in a local area. Also, the margin of error is always less than 20 cm^−1^ in the translation band, while it is much bigger in the stretching band. The PDOS curve has nine main peaks. The INS experiment by Li et al. reported 11 peaks in this band, of which five were detected between 115 and 198 cm^−1^ [6], whereas this work presents only one main peak.

On the basis of agreement with the experimental data, we analyzed the spectrum in terms of vibrational modes. Our previous study of ice Ic revealed two kinds of intrinsic translational modes in the ice lattice [7]. In the strong mode, the molecule vibrates together with four connecting H-bonds, whereas the weak mode involves only two oscillating H-bonds. Treating the two H-bonded molecules as a spring, a simple harmonic oscillator model yielded a strength ratio of $\sqrt{2}$.

Although the tetrahedral structure of ice II presents deformation under pressure to some extent, these two kinds of modes can also be distinguished in the translation region. For the mode at 318 cm^−1^ shown in Figure 2, all molecules vibrate along their H–O–H angular bisector, which yields a strong vibrational mode (please see Appendix A). The mode at 206 cm^−1^ is a two-bond mode, in which a molecule vibrates toward the two neighbors (please see Appendix A). We also found some kinds of cluster vibrational modes with much lower energy from 54 to 115 cm^−1^. As shown in Figure 2, the six-ring cluster vibrates in the same way at 115 cm^−1^, causing very little H-bond oscillation with the environment (please see Appendix A).

Based on this analysis, we classified the optic translational mode into three groups and fitted three PDOS curves, as shown in Figure 3. Note that the data of the fitted PDOS are from the modes in the BZ center, whereas the real PDOS covers the entire first BZ. Thus, some disagreements occur due to dispersions of of $\omega \left(q\right)$. Because the local tetrahedral structure was deformed under pressure, the distributions of the strong modes (in blue) extend downward to 200 cm^−1^. The weak modes constitute the red curve in between, and nine modes (in green) belong to cluster vibration or skeleton deformation, which have much lower energies. The obvious deviation at around 150 cm^−1^ is because the PDOS has a sharp peak, while the fitted curve has two. This may be because the dispersion of the lower modes present a big slope, which is close to 150 cm^−1^. Unlike many other ice phases, which have two sharp peaks in this region, the three main peaks above 150 cm^−1^ (PDOS) belong to strong and weak translational modes. This was shown to be the case in the INS experiments by Li [6,26].

The libration band contains 36 normal modes from 522 to 969 cm^−1^. Molecules have three types of vibration: rocking, or the rotation of the whole molecule around an axis perpendicular to the molecular plane; wagging, or the rotation of the molecule around an axis in the molecular plane, perpendicular to the bisector of the H–O–H angle; and twisting, or the rotation of the molecule around an axis coincident with the H–O–H angle bisector. Using the lowest frequency mode at 522 cm^−1^ as an example, the vibrational modes of molecules include a mixture of rocking and twisting, as shown in Figure 2. The mode at 733 cm^−1^ has the greatest intensity of IR activity but Raman inactivity in this band. In contrast, the mode at 774 cm^−1^ has the greatest Raman activity but IR inactivity, as shown in Figure 1.

Bertie and Whalley measured 15 IR peaks in the libration band (Table 1) [22], of which three peaks at 483, 498, and 1066 cm^−1^ were found to have no corresponding computational results. They also detected 11 Raman peaks in this band [25] that matched the normal modes very well. Only the mode at 969 cm^−1^ was detected experimentally. As seen in Figure 1, the intensity at 969 cm^−1^ is too small to be detected. For the PDOS curve, we may pick out 17 tiny peaks in the libration band. This trend is consistent with INS results, in that there are three sub-bands with four main peaks at 477, 548, 746, and 895 cm^−1^ [6].

The H–O–H bending band has 12 normal modes from 1659 to 1723 cm^−1^. These vibrations can be divided into in-phase and out-of-phase modes. The vibrations at 1659 and 1723 cm^−1^ that possess the lowest and highest energy in this band all present out-of-phase modes, as shown in Figure 4. An exception is the mode at 1703 cm^−1^, in which all molecules vibrate in-phase. Two peaks at 1690 and 1748 cm^−1^ were detected by IR absorption [22]. These correspond to the normal modes of 1680 and 1723 cm^−1^, according to the calculated IR intensities. For the Raman spectroscopy experiment, no peaks were detected in the bending area due to the weak intensity. The INS experiment detected a peak at 1669 cm^−1^ that corresponds to 1664 and 1699 cm^−1^ in PDOS [26].

The band ranging from 3193 to 3483 cm^−1^ is from intramolecular O–H stretching, which contains 24 normal modes. There are two vibrational modes for each molecule, known as symmetric and asymmetric stretching. In the modes of 3193 and 3198 cm^−1^, only symmetric stretching occurs. In the mode of 3393 cm^−1^, all molecules show asymmetric stretching. In the stretching band, many molecules present only one vibrating O–H bond, while the other keeps static, as shown in the 3483 cm^−1^ mode in Figure 4. This phenomenon can also be seen in other bands, such as that at 774 cm^−1^ in Figure 2. Whale et al. found in ice XV that some modes include the isolated vibration of only one O–H bond, whereas the other bond does not vibrate [30]. We also observed this phenomenon in ice XIV, XVI, XVII, and VII, but this did not occur in ice Ic and VIII. We regard this “isolated stretching mode” as a special case of asymmetric stretching and attribute it to local tetrahedral deformations by pressure and a hydrogen-disordered lattice.

Bertie et al. reported that six peaks of the IR spectrum [23] and seven peaks of the Raman scattering spectrum [25] in this band matched the active normal modes in Table 1. We also identified seven peaks in the PDOS spectrum, whereas the INS experiment recorded only one peak at 3387 cm^−1^.

## 4. Conclusions

In summary, using the first-principles density functional theory method, we present the theoretical Raman scattering, IR absorption, and INS spectra (PDOS) of ice II. The PDOS spectrum was used to assign the characteristic peaks in the INS spectrum because INS collects signals throughout the BZ without selection. The 105 optic normal modes in the BZ center could be compared with the Raman and IR spectra. Under this condition, the peaks recorded from photon scattering/absorption could be assigned individually according to the normal vibration mode.

The most valuable result was the identification of H-bond vibration modes in the translation band. Inspired by previous studies, we concluded that ice II also has two H-bond vibration modes, even though they show three main peaks, mainly due to the structural deformation of a local regular tetrahedron under pressure, similar to ice XIV. The lower energy regions also have cluster vibrational modes, which may overlap with acoustic phonons, as seen in the INS spectrum. We discussed the “isolated stretching mode” first discovered by Whale et al. in ice XV. In our work, such isolated stretching was found to be common in many vibrational modes. The tetrahedral structure in ice II showed great deformation, in that the bond length of the two H-bonds of one molecule in the six-ring cluster was 1.815 Å, whereas the other two were 1.955 and 2.057 Å. We attributed this phenomenon to local tetrahedral deformation.

## Figures and Tables

**Figure 1 molecules-24-03135-f001:**
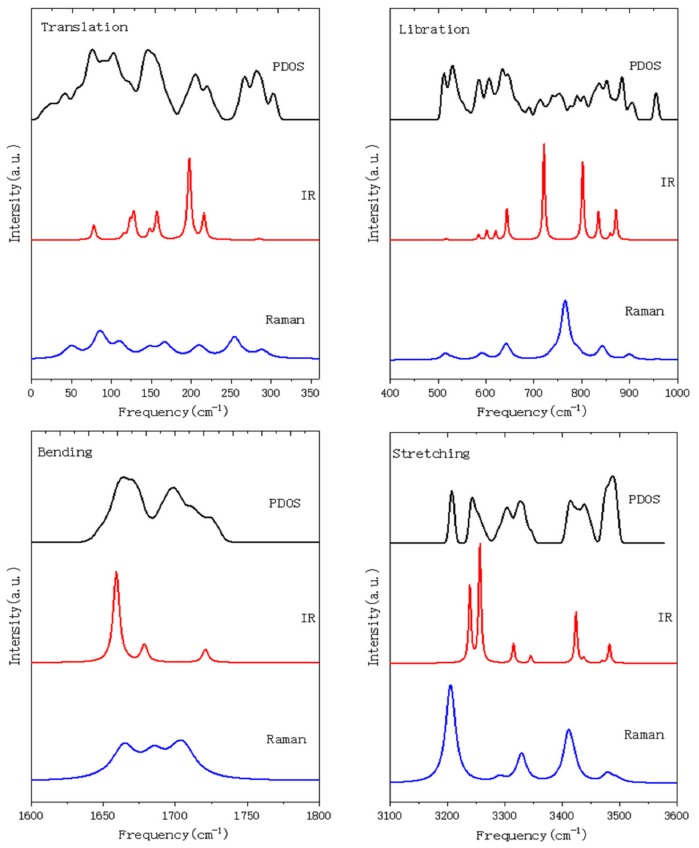
Simulated spectrum of ice II. The four images correspond to four vibration bands: translation, libration, bending, and stretching. From top to bottom: phonon density of states (PDOS), IR, and Raman spectra. Weak peaks have been amplified reasonably.

**Figure 2 molecules-24-03135-f002:**
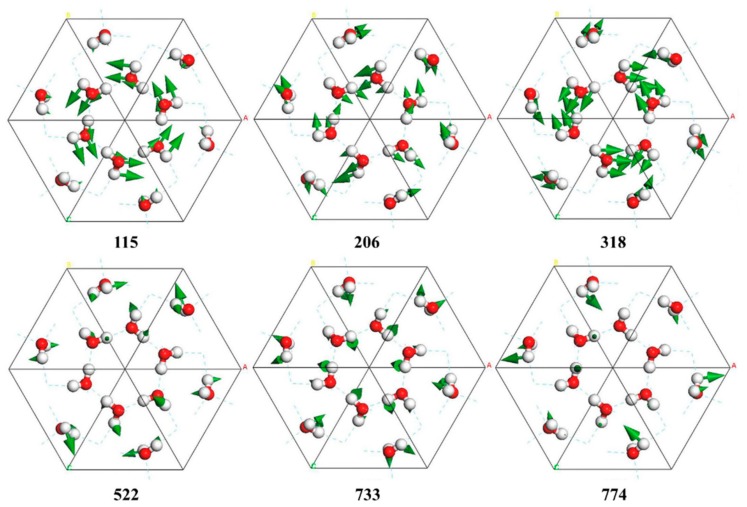
Top view of six normal modes in the translation band (115, 206, and 318 cm^−1^) and libration band (522, 733, and 774 cm^−1^). Green arrows represent vibration direction in sizes proportional to the vibration amplitude.

**Figure 3 molecules-24-03135-f003:**
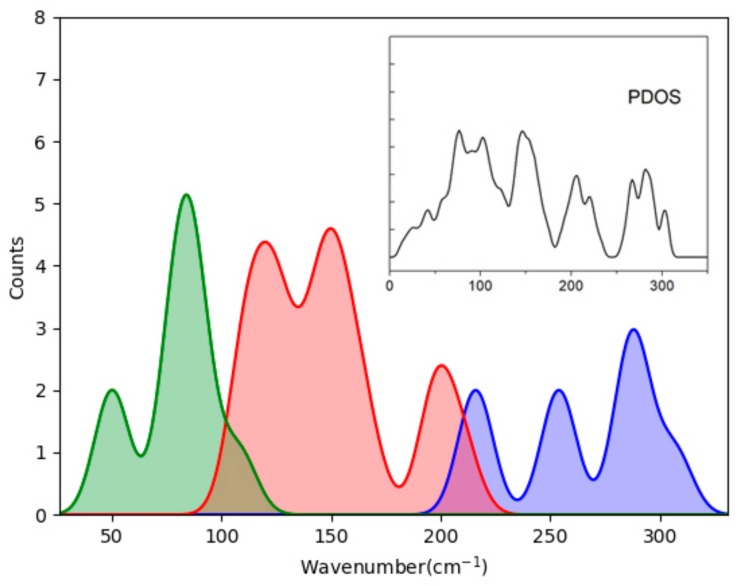
Fitted curves of four-bond modes (blue), two-bond modes (red), and cluster modes (green) of ice II in the translation band. The inset PDOS curve is shown for comparison. Note that the fitted curves are related to the Brillouin zone (BZ) center only, while the PDOS is integrated over the entire BZ.

**Figure 4 molecules-24-03135-f004:**
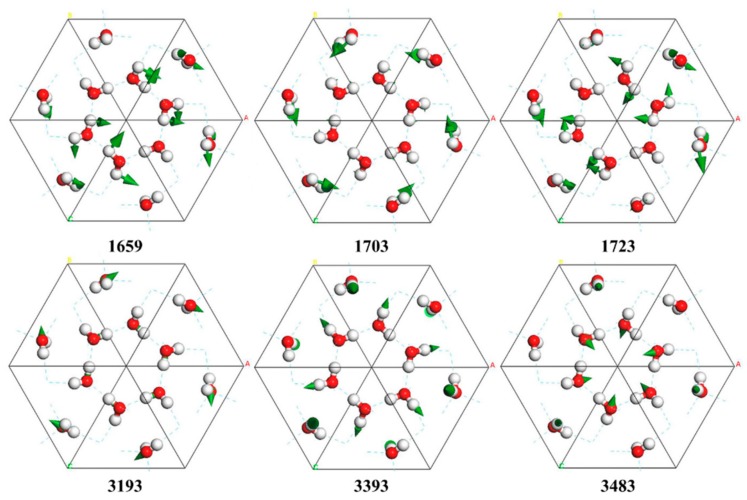
Top view of six normal modes in the bending band (1659, 1703, and 1723 cm^−1^) and stretching band (3193, 3393, and 3483 cm^−1^). The “isolated mode” can be seen in each stretching mode.

**Table 1 molecules-24-03135-t001:** Comparison of calculated results with inelastic neutron scattering (INS), IR, and Raman data. The main peaks of phonon density of states (PDOS) in the first column are compared against the INS spectrum. Normal modes (numbers in brackets indicate the degeneracies) are compared against experimental IR and Raman peaks. The values are expressed in cm^−1^.

PDOS	INS [6,26]	Normal Modes	IR [22,23]	Raman [24,25]
41	39	54 (2)		
76	83	82 (2)		
		86		
		90 (2)		72
		93		
102	96	115		
		117		104
		118 (2)		
		130 (2)	107	
		135		
		142		
147		154 (2)		152
	129	157	136	
	153	164 (2)	151	
	185	176		187
	198	206 (2)	186	
206		219		200
221	239	223 (2)	253	
267		262 (2)		262
282	294	293	295	
		298 (2)		268
302	327	318	335	322
513	477	522 (2)		489
		523 (2)	473	
			483	
			498	
		527	516	
529	548	538		495
585		597	533	
		599 (2)		573
		610		597
607		611	544	
635		630 (2)	593	
		654		617
646		655 (2)	642	
690		657 (2)		648
		660	660	
712		733	700	
738		752		685
753	746	774 (2)		715
790		804 (2)		775
805		815 (2)	745	
838		844 (2)	800	
851	895	853 (2)		845
		871	835	
884		881 (2)	960	
904		909		950
954		969		
			1066	
1664	1669	1659 (2)		
		1661		
		1665 (2)		
		1680	1690	
		1686 (2)		
1699		1703		
		1708		
		1723 (2)	1748	
			2220	
			2300	
3207		3193		3189
		3198		3225
3242	3387	3227 (2)	3000	
		3245 (2)	3225	
		3279 (2)		
3303		3304	3280	
3327		3321 (2)		3270
		3335	3390	
3414		3393		3306
3438		3405 (2)		3340
		3412	3470	
		3425 (2)		
		3456		
3488		3466 (2)		3400
		3469 (2)	3500	
		3483		3465

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
