# Peer review of "DFT Investigations of the Vibrational Spectra and Translational Modes of Ice II"

_molecules, 2019, doi:10.3390/molecules24173135_

Round 1
Reviewer 1 Report
Authors improved the quality of the manuscript. The revised version is acceptable for publication in Molecules.
Author Response
Many thanks.
Reviewer 2 Report
The response to my first comment is very formal while to the second one does not answer my question at all. At least, change “much big” to “much bigger”. It seems the authors are able neither to draw general conclusions from their multiple studies of vibrational spectra of different ice phases nor to present their data in a way attractive to a wide audience. I hope it can be somehow useful for experts in the field. Not being such an expert myself, I cannot give any specific recommendation how to improve the article.
Author Response
1.The response to my first comment is very formal while to the second one does not answer my question at all. At least, change “much big” to “much bigger”. It seems the authors are able neither to draw general conclusions from their multiple studies of vibrational spectra of different ice phases nor to present their data in a way attractive to a wide audience. I hope it can be somehow useful for experts in the field. Not being such an expert myself, I cannot give any specific recommendation how to improve the article. Response: We thank the reviewer's professional comments. The grammar error was amended to "much bigger". As for the theoretical analysis of the translational band, we will present the formula derivation in another paper later on the simulation of ice Ih.Reviewer 3 Report
Please, see the attached file

Author Response
1. [Page 1, lines 16-17] in the abstract. The reference to Whale et al. on ice XV has been modified with respect to that contained in the previous version of the manuscript, but, in my view, it is not yet perfectly clear . I have interpreted the sentence as follows: “ Whale et al. found in ice XV that some intra - molecular vibrational modes include many isolated - molecule stretches of only one O – H bond, whereas the other O – H bond does not vibrate. ” Could the authors confirm my interpretation? Response: Thank you for your advice. We replaced this sentence as recommendation from line 16 to 18. 2. [Page 6, lines 220-221] Although it is already mentioned in the main text, I think it is important to report in the caption of Fig. 3 the fact that the fitted curves are related to the BZ center only, while the PDOS is averaged over the entire BZ. Response: OK, we added one sentence in the caption of figure 3 as below: "Note that the fitted curves are related to the BZ center only, while the PDOS is integrated over the entire BZ." 3. [all pages 1-10] Since the manuscript text has been modified in a substantial way, a number of new paragraphs have been included. These are generally understandable from the scientific point of view, but, surely, they still need an accurate grammar check. Response: Thanks. We re-checked the text and amended some bugs.Reviewer 4 Report
I would prefer to read with subtitles, but the rewriting was enough to make the read more fluent
Author Response
Thanks for your advice. Although we discussed the four parts of the vibration spectrum, we focused on the molecular vibrations in the translation region. Since the other three parts were very short, we do not need to set subtitles for each part.
This manuscript is a resubmission of an earlier submission. The following is a list of the peer review reports and author responses from that submission.
Round 1
Reviewer 1 Report
This manuscript presents DFT investigations of the vibrational spectra and translational modes of ice II by Zhang and co-workers. Authors used CASTEP code with PBE functional for the DFT calculations. The reliability of the functional was not justified. Authors highlighted their experience which is not a proper justification. Still, this is not a solid state calculation in order to compare with experiments. Authors also analyzed the vibrational frequencies but not clearly stated in the abstract. The importance of this work is not revealed / justified. It seems to be me that some calculations were done and compared with experiments. Therefore, I do not recommend publication of this manuscript in Molecules.
See the comments below:
The units for the data in Table 1 are not given. It is not easy to quickly understand the data in Table 1. It is better to separate IR and Raman modes.
Abstract is not well written. Some statements in the abstract should go to the introduction section.
Computational details section is very brief, and it is difficult for others to follow and reproduce. Some information is missing.
Reviewer 2 Report
IR, Raman and INS spectra of ice II have been simulated using the phonon density of states approach. The results of this simulation has been used in order to interpret available experimental data in terms of vibrational modes. The authors have collected a large number of literature experimental data. Thus, the reported analysis can be useful for experts in the field. Indeed, a number of similar papers have been published by the authors about other forms of ice. Some of them are cited moderately often.
The authors do not discuss the margin of errors of their calculations. They do not explain why the deviations between the calculated normal modes and the experimental frequencies vary in a non-systematic way. Videos from the Supplementary Materials help somewhat to understand the results but cannot compensate for the absence of their general discussion. Therefore, in my opinion, the manuscript can be accepted for publication after a minor text revision. There are quite a few errors in grammar through out the manuscript.
Reviewer 3 Report
Report on manuscript “molecules-562986” “DFT investigations of the vibrational spectra and translation modes of ice II” The manuscript “DFT investigations of the vibrational spectra and translation modes of ice II” by Jing- Wen Cao et al. submitted for publication on “Molecules” deals with the lattice dynamics of ice II, investigated using DFT ab initio simulations through the well-known computational code CASTEP. The work reports normal mode vibrations at the Brillouin zone (BZ) center, as well as the full phonon density of states (PDOS) integrated over the complete BZ. Comparisons with infrared (IR), Raman and incoherent inelastic neutron scattering (INS) data are also provided. A physical interpretation of some selected spectral features is finally suggested, pointing out differences and similarities with other ice forms. The work is scientifically original and very sound, so it surely deserves to be published. However, there are still some issues to be addressed and a few points to be clarified before being accepted in its final version: 1) [Page 1, lines 14-15] in the abstract. The reference to ice XV is not clear. I have interpreted the sentence as: “(…) whereas others do not vibrate, like in ice XV”. Is it correct? 2) [Page 1, line 39] the second mention to Ref. [10] seems incorrect. I think it should be Ref. [11]. The authors should check. 3) [Page 2, line 45] “was” should be replaced by “were”. 4) [Page 2, lines 52-57] the “Computational Methods” section is sketchy. Could the authors add some other details? Why did they use GGA-RPBE instead of, for example, GGA-PBE? Are van- der-Waals interactions (e.g. through TS or Grimme schemes) somehow included? Are the used pseudo-potentials “ultra-soft” or “norm-conserving”? Have the phonons been calculated via “finite displacement” or “density-functional perturbation theory method”? 5) [Page 2, line 62] The authors claim that INS data are in direct proportion to PDOS. Strictly speaking, this is not completely true for two main reasons: (a) the Debye-Waller factor as well as the Bose population factor are included in the INS spectra, but not in the PDOS definition; (b) rather than the full density of phonon states, owing to the large H cross-section for thermal neutrons, INS spectra contain the so-called “H-projected density of phonon states” (H-DOPS). A quantitative relationship between INS spectrum and H-DOPS can be found, for example, in Ref. [19] in Eqs. (1) and (2). It is surely correct to point out qualitative similarities between INS spectra and PDOS, but some caution is surely needed here, as some bands in the PDOS could turn out to be rather weak in the H-PDOS. 6) [Page 5, line 88] J.-C. Li and coworkers report 18 peaks in their INS results, not only 11. The authors should explain why they mentioned only 11 peaks. 7) [Page 6, line 91] “mode” should be replaced by “modes”. 8) [Page 6, line 102] “its” should be replaced by “their”. 9) [Page 7, line 134] “was been” should be replaced by “has been”. 10) [Page 8, line 159] the sentence “(…) this phenomenon in ice XIV, VII, and so on, but (…) ” is not clear and should be modified listing (in an ordered way) all the ice forms in which this phenomenon occurs, and all those in which this phenomenon does not occur. 11) [Page 8, line 162] the sentence “It was reported that six peaks (…)” is not clear and should be modified. I have interpreted it as “We have reported six peaks (…)”. Is it correct? 12) [Page 9, line 172] the word “scattering” is appropriate for Raman spectra only, not for the IR ones. The authors should replace it with “scattering/absorption”. 13) [Page 9, line 177] the sentence “(…) tetrahedron under pressure similar to ice XIV” is not clear and should be modified. I have interpreted it as. “(…) tetrahedron under pressure, similarly to ice XIV”. Is it correct? 14) [Page 9, lines 196-200] References [1] and [3] should be swapped. 15) [Page 9, lines 201-202] Reference [4] looks incorrect. It should read Tammann, G. Ueber die Grenzen des festen Zustandes IV. Annalen Der Physik 1900, 307(5), 1–31.Reviewer 4 Report
I have reviewed the Manuscript molecules-562986, Titled DFT investigations of the vibrational spectra and translational modes of ice II.
These are my comments:
The topic is interesting but the authors should explain in the manuscript the reason to research the translation modes particularly on ice II. A slight definition of motivation should be included.
In the computational methods section, the authors do not explain the reason to define de pressure at 0.3 GPa. I think that it is very important to define this particular decision considering that PDOS can show very different behavior according to pressure.
If the authors do not explain the reason to define the pressure, I think that the results would be more important and transcendent with calculations at different pressures.
Even when the results are interesting, the authors could organize it. The lecture is really confusing. They can include subtitles and organize paragraphs.
As an example, line 65 include this phrase: "The related typical normal modes are discussed later in this paper". But It was difficult to find the discussion.
The work is well developed but not well written.
For this reason, I do not recommend the paper in the present form for publication in Molecules.